# REVISITING FEW-SAMPLE BERT FINE-TUNING

**Tianyi Zhang**[*△§]    **Felix Wu**[*†]    **Arzoo Katiyar**[△◇]    **Kilian Q. Weinberger**[†‡]    **Yoav Artzi**[†‡]

[†]ASAPP Inc.    [§]Stanford University    [◇]Penn State University    [‡]Cornell University
tz58@stanford.edu  {fwu, kweinberger, yoav}@asapp.com  arzoo@psu.edu

## ABSTRACT

This paper is a study of fine-tuning of BERT contextual representations, with focus on commonly observed instabilities in few-sample scenarios. We identify several factors that cause this instability: the common use of a non-standard optimization method with biased gradient estimation; the limited applicability of significant parts of the BERT network for down-stream tasks; and the prevalent practice of using a pre-determined, and small number of training iterations. We empirically test the impact of these factors, and identify alternative practices that resolve the commonly observed instability of the process. In light of these observations, we re-visit recently proposed methods to improve few-sample fine-tuning with BERT and re-evaluate their effectiveness. Generally, we observe the impact of these methods diminishes significantly with our modified process.

## 1    INTRODUCTION

Fine-tuning self-supervised pre-trained models has significantly boosted state-of-the-art performance on natural language processing (NLP) tasks (Liu, 2019; Yang et al., 2019a; Wadden et al., 2019; Zhu et al., 2020; Guu et al., 2020). One of the most effective models for this process is BERT (Devlin et al., 2019). However, despite significant success, fine-tuning remains unstable, especially when using the large variant of BERT (BERT$_{\text{Large}}$) on small datasets, where pre-training stands to provide the most significant benefit. Identical learning processes with different random seeds often result in significantly different and sometimes degenerate models following fine-tuning, even though only a few, seemingly insignificant aspects of the learning process are impacted by the random seed (Phang et al., 2018; Lee et al., 2020; Dodge et al., 2020).[1] As a result, practitioners resort to multiple random trials for model selection. This increases model deployment costs and time, and makes scientific comparison challenging (Dodge et al., 2020).

This paper is a study of different aspects of the few-sample fine-tuning optimization process. Our goal is to better understand the impact of common choices with regard to the optimization algorithm, model initialization, and the number of fine-tuning training iterations. We identify suboptimalities in common community practices: the use of a non-standard optimizer introduces bias in the gradient estimation; the top layers of the pre-trained BERT model provide a bad initialization point for fine-tuning; and the use of a pre-determined , but commonly adopted number of training iterations hurts convergence. We study these issues and their remedies through experiments on multiple common benchmarks, focusing on few-sample fine-tuning scenarios.

Once these suboptimal practices are addressed, we observe that degenerate runs are eliminated and performance becomes much more stable. This makes it unnecessary to execute numerous random restarts as proposed in Dodge et al. (2020). Our experiments show the remedies we experiment with for each issue have overlapping effect. For example, allocating more training iterations can eventually compensate for using the non-standard biased optimizer, even though the combination of a bias-corrected optimizer and re-initializing some of the pre-trained model parameters can reduce fine-tuning computational costs. This empirically highlights how different aspects of fine-tuning influence the stability of the process, at times in a similar manner. In the light of our observations, we re-evaluate several techniques (Phang et al., 2018; Lee et al., 2020; Howard & Ruder, 2018) that

---

[*]Equal contribution, [△] Work done at ASAPP.

[1]Fine-tuning  instability  is  also  receiving  significant  practitioner  attention.    For  example: https://github.com/zihangdai/xlnet/issues/96 and https://github.com/huggingface/transformers/issues/265.

were recently proposed to increase few-sample fine-tuning stability and show a significant decrease in their impact. Our work furthers the empirical understanding of the fine-tuning process, and the optimization practices we outline identify impactful avenues for the development of future methods.

## 2 BACKGROUND AND RELATED WORK

**BERT** The Bidirectional Encoder Representations from Transformers (BERT; Devlin et al., 2019) model is a Transformer encoder (Vaswani et al., 2017) trained on raw text using masked language modeling and next-sentence prediction objectives. It generates an embedding vector contextualized through a stack of Transformer blocks for each input token. BERT prepends a special `[CLS]` token to the input sentence or sentence pairs. The embedding of this token is used as a summary token for the input for classification tasks. This embedding is computed with an additional fully-connected layer with a `tanh` non-linearity, commonly referred to as the *pooler*, to aggregate the information for the `[CLS]` embedding.

**Fine-tuning** The common approach for using the pre-trained BERT model is to replace the original output layer with a new task-specific layer and fine-tune the complete model. This includes learning the new output layer parameters and modifying all the original weights, including the weights of word embeddings, Transformer blocks, and the pooler. For example, for sentence-level classification, an added linear classifier projects the `[CLS]` embedding to an unnormalized probability vector over the output classes. This process introduces two sources of randomness: the weight initialization of the new output layer and the data order in the stochastic fine-tuning optimization. Existing work (Phang et al., 2018; Lee et al., 2020; Dodge et al., 2020) shows that these seemingly benign factors can influence the results significantly, especially on small datasets (i.e., $< 10K$ examples). Consequently, practitioners often conduct many random trials of fine-tuning and pick the best model based on validation performance (Devlin et al., 2019).

**Fine-tuning Instability** The instability of the BERT fine-tuning process has been known since its introduction (Devlin et al., 2019), and various methods have been proposed to address it. Phang et al. (2018) show that fine-tuning the pre-trained model on a large intermediate task stabilizes later fine-tuning on small datasets. Lee et al. (2020) introduce a new regularization method to constrain the fine-tuned model to stay close to the pre-trained weights and show that it stabilizes fine-tuning. Dodge et al. (2020) propose an early stopping method to efficiently filter out random seeds likely to lead to bad performance. Concurrently to our work, Mosbach et al. (2020) also show that BERTADAM leads to instability during fine-tuning. Our experiments studying the effect of training longer are related to previous work studying this question in the context of training models from scratch (Popel & Bojar, 2018; Nakkiran et al., 2019).

**BERT Representation Transferability** BERT pre-trained representations have been widely studied using probing methods showing that the pre-trained features from intermediate layers are more transferable (Tenney et al., 2019b;a; Liu et al., 2019a; Hewitt & Manning, 2019; Hewitt & Liang, 2019) or applicable (Zhang et al., 2020) to new tasks than features from later layers, which change more after fine-tuning (Peters et al., 2019; Merchant et al., 2020). Our work is inspired by these findings, but focuses on studying how the pre-trained weights influence the fine-tuning process. Li et al. (2020) propose to re-initialize the final fully-connected layer of a ConvNet and show performance gain for image classification.[2] Concurrent to our work, Tamkin et al. (2020) adopt a similar methodology of weight re-initialization (Section 5) to study the transferability of BERT. In contrast to our study, their work emphasizes pinpointing the layers that contribute the most in transfer learning, and the relation between probing performance and transferability.

## 3 EXPERIMENTAL METHODOLOGY

**Data** We follow the data setup of previous studies (Lee et al., 2020; Phang et al., 2018; Dodge et al., 2020) to study few-sample fine-tuning using eight datasets from the GLUE benchmark (Wang et al., 2019b). The datasets cover four tasks: natural language inference (RTE, QNLI, MNLI), paraphrase detection (MRPC, QQP), sentiment classification (SST-2), and linguistic acceptability (CoLA). Appendix A provides dataset statistics and a description of each dataset. We primarily

---

[2]This concurrent work was published shortly after our study was posted.

---

**Algorithm 1:** the ADAM pseudocode adapted from Kingma & Ba (2014), and provided for reference. $g_t^2$ denotes the elementwise square $g_t \odot g_t$. $\beta_1$ and $\beta_2$ to the power $t$ are denoted as $\beta_1^t$ $\beta_2^t$. All operations on vectors are element-wise. The suggested hyperparameter values according to Kingma & Ba (2014) are: $\alpha = 0.001$, $\beta_1 = 0.9$, $\beta_2 = 0.999$, and $\epsilon = 10^{-8}$. BERTADAM (Devlin et al., 2019) omits the bias correction (lines 9–10), and treats $m_t$ and $v_t$ as $\widehat{m}_t$ and $\widehat{v}_t$ in line 11.

---

**Require:** $\alpha$: learning rate; $\beta_1, \beta_2 \in [0, 1)$: exponential decay rates for the moment estimates; $f(\theta)$: stochastic objective function with parameters $\theta$; $\theta_0$: initial parameter vector; $\lambda \in [0, 1)$: decoupled weight decay.

1: $m_0 \leftarrow 0$ (Initialize first moment vector)
2: $v_0 \leftarrow 0$ (Initialize second moment vector)
3: $t \leftarrow 0$ (Initialize timestep)
4: **while** $\theta_t$ not converged **do**
5:    $t \leftarrow t + 1$
6:    $g_t \leftarrow \nabla_\theta f_t(\theta_{t-1})$ (Get gradients w.r.t. stochastic objective at timestep $t$)
7:    $m_t \leftarrow \beta_1 \cdot m_{t-1} + (1 - \beta_1) \cdot g_t$ (Update biased first moment estimate)
8:    $v_t \leftarrow \beta_2 \cdot v_{t-1} + (1 - \beta_2) \cdot g_t^2$ (Update biased second raw moment estimate)
9:    $\widehat{m}_t \leftarrow m_t/(1 - \beta_1^t)$ (Compute bias-corrected first moment estimate)
10:    $\widehat{v}_t \leftarrow v_t/(1 - \beta_2^t)$ (Compute bias-corrected second raw moment estimate)
11:    $\theta_t \leftarrow \theta_{t-1} - \alpha \cdot \widehat{m}_t/(\sqrt{\widehat{v}_t} + \epsilon)$ (Update parameters)
12: **end while**
13: **return** $\theta_t$ (Resulting parameters)

---

focus on four datasets (RTE, MRPC, STS-B, CoLA) that have fewer than 10k training samples, because BERT fine-tuning on these datasets is known to be unstable (Devlin et al., 2019). We also complement our study by downsampling all eight datasets to 1k training examples following Phang et al. (2018). While previous studies (Lee et al., 2020; Phang et al., 2018; Dodge et al., 2020) focus on the validation performance, we split held-out test sets for our study.[3] For RTE, MRPC, STS-B, and CoLA, we divide the original validation set in half, using one half for validation and the other for test. For the other four larger datasets, we only study the downsampled versions, and split additional 1k samples from the training set as our validation data and test on the original validation set.

**Experimental Setup** Unless noted otherwise, we follow the hyperparameter setup of Lee et al. (2020). We fine-tune the uncased, 24-layer BERT$_{\text{Large}}$ model with batch size 32, dropout 0.1, and peak learning rate $2 \times 10^{-5}$ for three epochs. We clip the gradients to have a maximum norm of 1. We apply linear learning rate warm-up during the first 10% of the updates followed by a linear decay. We use mixed precision training using Apex[4] to speed up experiments. We show that mixed precision training does not affect fine-tuning performance in Appendix C. We evaluate ten times on the validation set during training and perform early stopping. We fine-tune with 20 random seeds to compare different settings.

## 4 OPTIMIZATION ALGORITHM: DEBIASING OMISSION IN BERTADAM

The most commonly used optimizer for fine-tuning BERT is BERTADAM, a modified version of the ADAM first-order stochastic optimization method. It differs from the original ADAM algorithm (Kingma & Ba, 2014) in omitting a bias correction step. This change was introduced by Devlin et al. (2019), and subsequently made its way into common open source libraries, including the official implementation,[5] huggingface's Transformers (Wolf et al., 2019),[6] AllenNLP (Gardner et al., 2018), GluonNLP (Guo et al., 2019), jiant (Wang et al., 2019c), MT-DNN (Liu et al., 2020), and FARM.[7] As a result, this non-standard implementation is widely used in both industry and research (Wang et al., 2019a; Phang et al., 2018; Lee et al., 2020; Dodge et al., 2020; Sun et al., 2019; Clark et al., 2020; Lan et al., 2020; Houlsby et al., 2019; Stickland & Murray, 2019; Liu et al., 2019b). We observe that the bias correction omission influences the learning rate, especially early in the fine-tuning process, and is one of the primary reasons for instability in fine-tuning BERT (Devlin et al., 2019; Phang et al., 2018; Lee et al., 2020; Dodge et al., 2020).

Algorithm 1 shows the ADAM algorithm, and highlights the omitted line in the non-standard BERTADAM implementation. At each optimization step (lines 4–11), ADAM computes the exponen-

---

[3]The original test sets are not publicly available.
[4]https://github.com/NVIDIA/apex
[5]https://github.com/google-research/bert/blob/f39e881/optimization.py#L108-L157
[6]The default was changed from BERTADAM to debiased ADAM in commit ec07cf5a on July 11, 2019.
[7]https://github.com/deepset-ai/FARM

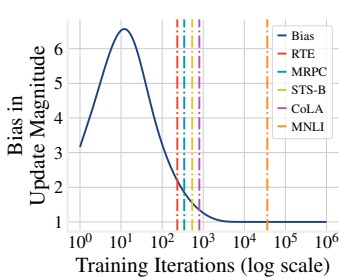

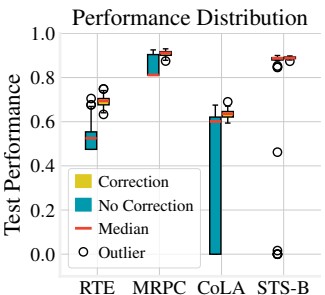

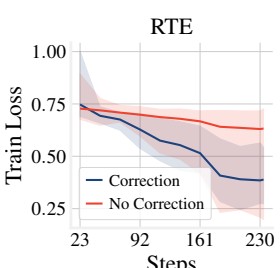

Figure 1: Bias in the ADAM update as a function of training iterations. Vertical lines indicate the typical number of iterations used to fine-tune BERT on four small datasets and one large dataset (MNLI). Small datasets use fewer iterations and are most affected.

Figure 2: Performance distribution box plot across 50 random trials and the four datasets with and without ADAM bias correction. Bias correction reduces the variance of fine-tuning results by a large margin.

Figure 3: Mean (solid lines) and range (shaded region) of training loss during fine-tuning BERT, across 50 random trials. Bias correction speeds up convergence and shrinks the range of training loss.

tial moving average of the gradients ($m_t$) and the squared gradients ($v_t$), where $\beta_1, \beta_2$ parameterize the averaging (lines 7–8). Because ADAM initializes $m_t$ and $v_t$ to 0 and sets exponential decay rates $\beta_1$ and $\beta_2$ close to 1, the estimates of $m_t$ and $v_t$ are heavily biased towards 0 early during learning when $t$ is small. Kingma & Ba (2014) computes the ratio between the biased and the unbiased estimates of $m_t$ and $v_t$ as $(1 - \beta_1^t)$ and $(1 - \beta_2^t)$. This ratio is independent of the training data. The model parameters $\theta$ are updated in the direction of the averaged gradient $m_t$ divided by the square root of the second moment $\sqrt{v_t}$ (line 11). BERTADAM omits the debiasing (lines 9–10), and directly uses the biased estimates in the parameters update.

Figure 1 shows the ratio $\frac{\hat{m}_t}{\sqrt{\hat{v}_t}}$ between the update using the biased and the unbiased estimation as a function of training iterations. The bias is relatively high early during learning, indicating overestimation. It eventually converges to one, suggesting that when training for sufficient iterations, the estimation bias will have negligible effect.[8] Therefore, the bias ratio term is most important early during learning to counteract the overestimation of $m_t$ and $v_t$ during early iterations. In practice, ADAM adaptively re-scales the learning rate by $\frac{\sqrt{1-\beta_2^t}}{1-\beta_1^t}$. This correction is crucial for BERT fine-tuning on small datasets with fewer than 10k training samples because they are typically fine-tuned with less than 1k iterations (Devlin et al., 2019). The figure shows the number of training iterations for RTE, MRPC, STS-B, CoLA, and MNLI. MNLI is the only one of this set with a large number of supervised training examples. For small datasets, the bias ratio is significantly higher than one for the entire fine-tuning process, implying that these datasets suffer heavily from overestimation in the update magnitude. In comparison, for MNLI, the majority of fine-tuning occurs in the region where the bias ratio has converged to one. This explains why fine-tuning on MNLI is known to be relatively stable (Devlin et al., 2019).

We evaluate the importance of the debiasing step empirically by fine-tuning BERT with both BERTADAM and the debiased ADAM[9] for 50 random seeds on RTE, MRPC, STS-B, and CoLA. Figure 2 summarizes the performance distribution. The bias correction significantly reduces the performance variance across different random trials and the four datasets. Without the bias correction we observe many degenerate runs, where fine-tuned models fail to outperform the random baseline. For example, on RTE, 48% of fine-tuning runs have an accuracy less than 55%, which is close to random guessing. Figure 3 further illustrates this difference by plotting the mean and the range of training loss during fine-tuning across different random trials on RTE. Figure 11 in Appendix F shows similar plots for MRPC, STS-B, and CoLA. The biased BERTADAM consistently leads to worse averaged training loss, and on all datasets to higher maximum training loss. This indicates models trained with BERTAdam are underfitting and the root of instability lies in optimization.

---

[8]Our experiments on the completely MNLI dataset confirm using the unbiased estimation does not improve nor degrade performance for large datasets (Appendix D).

[9]We use the PyTorch ADAM implementation https://pytorch.org/docs/1.4.0/_modules/torch/optim/adamw.html.

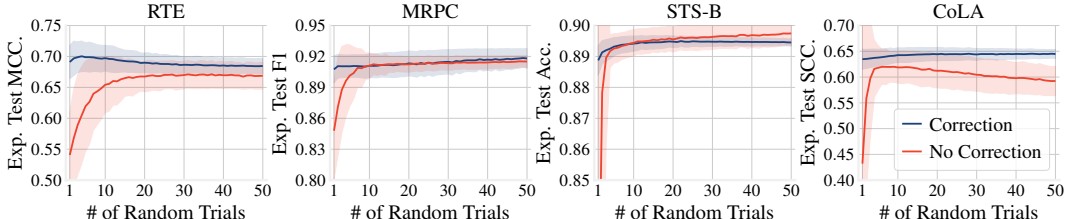

Figure 4: Expected test performance (solid lines) with standard deviation (shaded region) over the number of random trials allocated for fine-tuning BERT. With bias correction, we reliably achieve good results with few (i.e., 5 or 10) random trials.

| Dataset | RTE | | MRPC | | STS-B | | CoLA | |
|---|---|---|---|---|---|---|---|---|
| | 3 Epochs | Longer | 3 Epochs | Longer | 3 Epochs | Longer | 3 Epochs | Longer |
| Standard | $69.5 \pm 2.5$ | $72.3 \pm 1.9$ | $90.8 \pm 1.3$ | $90.5 \pm 1.5$ | $89.0 \pm 0.6$ | $89.6 \pm 0.3$ | $63.0 \pm 1.5$ | $62.4 \pm 1.7$ |
| Re-init | $72.6 \pm 1.6$ | $73.1 \pm 1.3$ | $91.4 \pm 0.8$ | $91.0 \pm 0.4$ | $89.4 \pm 0.2$ | $89.9 \pm 0.1$ | $63.9 \pm 1.9$ | $61.9 \pm 2.3$ |
| Dataset | RTE (1k) | | MRPC (1k) | | STS-B (1k) | | CoLA (1k) | |
| | 3 Epochs | Longer | 3 Epochs | Longer | 3 Epochs | Longer | 3 Epochs | Longer |
| Standard | $62.5 \pm 2.8$ | $65.2 \pm 2.1$ | $80.5 \pm 3.3$ | $83.8 \pm 2.1$ | $84.7 \pm 1.4$ | $88.0 \pm 0.4$ | $45.9 \pm 1.6$ | $48.8 \pm 1.4$ |
| Re-init | $65.6 \pm 2.0$ | $65.8 \pm 1.7$ | $84.6 \pm 1.6$ | $86.0 \pm 1.2$ | $87.2 \pm 0.4$ | $88.4 \pm 0.2$ | $47.6 \pm 1.8$ | $48.4 \pm 2.1$ |
| Dataset | SST (1k) | | QNLI (1k) | | QQP (1k) | | MNLI (1k) | |
| | 3 Epochs | Longer | 3 Epochs | Longer | 3 Epochs | Longer | 3 Epochs | Longer |
| Standard | $89.7 \pm 1.5$ | $90.9 \pm 0.5$ | $78.6 \pm 2.0$ | $81.4 \pm 0.9$ | $74.0 \pm 2.7$ | $77.4 \pm 0.8$ | $52.2 \pm 4.2$ | $67.5 \pm 1.1$ |
| Re-init | $90.8 \pm 0.4$ | $91.2 \pm 0.5$ | $81.9 \pm 0.5$ | $82.1 \pm 0.3$ | $77.2 \pm 0.7$ | $77.6 \pm 0.6$ | $66.4 \pm 0.6$ | $68.8 \pm 0.5$ |

Table 1: Mean test performance and standard deviation. We compare fine-tuning with the complete BERT model (Standard) and fine-tuning with the partially re-initialized BERT (Re-init). We show results of fine-tuning for 3 epochs and for longer training (Sec 6). We underline and highlight in blue the best and number statistically equivalent to it among each group of 4 numbers. We use a one-tailed Student's $t$-test and reject the null hypothesis when $p < 0.05$.

We simulate a realistic setting of multiple random trials following Dodge et al. (2020). We use bootstrapping for the simulation: given the 50 fine-tuned models we trained, we sample models with replacement, perform model selection on the validation set, and record the test results; we repeat this process 1k times to estimate mean and variance. Figure 4 shows the simulated test results as a function of the number of random trials. Appendix E provides the same plots for validation performance. Using the debiased ADAM we can reliably achieve good results using fewer random trials; the difference in expected performance is especially pronounced when we perform less than 10 trials. Whereas the expected validation performance monotonically improves with more random trials (Dodge et al., 2020), the expected test performance deteriorates when we perform too many random trials because the model selection process potentially overfits the validation set. Based on these observations, we recommend performing a moderate number of random trials (i.e., 5 or 10).

## 5 INITIALIZATION: RE-INITIALIZING BERT PRE-TRAINED LAYERS

The initial values of network parameters have significant impact on the process of training deep neural networks, and various methods exist for careful initialization (Glorot & Bengio, 2010; He et al., 2015; Zhang et al., 2019; Radford et al., 2019; Dauphin & Schoenholz, 2019). During fine-tuning, the BERT parameters take the role of the initialization point for the fine-tuning optimization process, while also capturing the information transferred from pre-training. The common approach for BERT fine-tuning is to initialize all layers except one specialized output layer with the pre-trained weights. We study the value of transferring all the layers in contrast to simply ignoring the information learned in some layers. This is motivated by object recognition transfer learning results showing that lower pre-trained layers learn more general features while higher layers closer to the output specialize more to the pre-training tasks (Yosinski et al., 2014). Existing methods using BERT show that using the complete network is not always the most effective choice, as we discuss in Section 2. Our empirical results further confirm this: we observe that transferring the top pre-trained layers slows down learning and hurts performance.

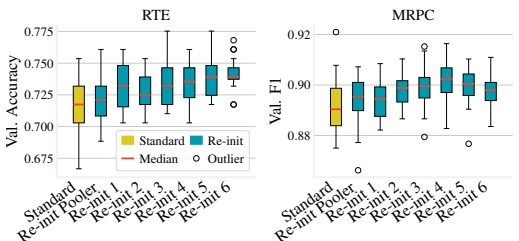
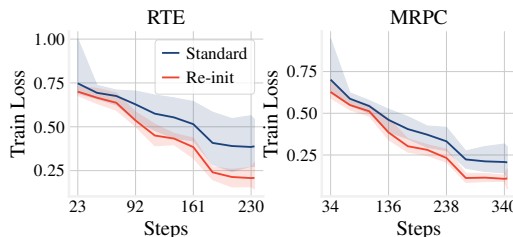

Figure 5: Validation performance distribution of re-initializing different number of layers of the BERT model.

Figure 6: Mean (solid lines) and Range (shaded region) of training loss during fine-tuning BERT, across 20 random trials. Re-init leads to faster convergence and shrinks the range.

We test the transferability of the top layers using a simple ablation study. Instead of using the pre-trained weights for all layers, we re-initialize the pooler layers and the top $L \in \mathbb{N}$ BERT Transformer blocks using the original BERT initialization, $\mathcal{N}(0, 0.02^2)$. We compare two settings: (a) standard fine-tuning with BERT, and (b) Re-init fine-tuning of BERT. We evaluate Re-init by selecting $L \in \{1, \dots, 6\}$ based on mean validation performance. All experiments use the debiased ADAM (Section 4) with 20 random seeds.

**Re-init Impact on Performance** Table 1 shows our results on all the datasets from Section 3. We show results for the common setting of using 3 epochs, and also for longer training, which we discuss and study in Section 6. Re-init consistently improves mean performance on all the datasets, showing that not all layers are beneficial for transferring. It usually also decreases the variance across all datasets. Appendix F shows similar benefits for pre-trained models other than BERT.

**Sensitivity to Number of Layers Re-initialized** Figure 5 shows the effect of the choice of $L$, the number of blocks we re-initialize, on RTE and MRPC. Figure 13 in Appendix F shows similar plots for the rest of the datasets. We observe more significant improvement in the worst-case performance than the best performance, suggesting that Re-init is more robust to unfavorable random seed. We already see improvements when only the pooler layer is re-initialized. Re-initializing further layers helps more. For larger $L$ though, the performance plateaus and even decreases as re-initialize pre-trained layers with general important features. The best $L$ varies across datasets.

**Effect on Convergence and Parameter Change** Figure 6 shows the training loss for both the standard fine-tuning and Re-init on RTE and MRPC. Figure 13, Appendix F shows the training loss for all other datasets. Re-init leads to faster convergence. We study the weights of different Transformer blocks. For each block, we concatenate all parameters and record the $L2$ distance between these parameters and their initialized values during fine-tuning. Figure 7 plots the $L2$ distance for four different transformer blocks as a function of training steps on RTE, and Figures 15–18 in Appendix F show all transformer blocks on four datasets. In general, Re-init decreases the $L2$ distance to initialization for top Transformer blocks (i.e., 18–24). Re-initializing more layers leads to a larger reduction, indicating that Re-init decreases the fine-tuning workload. The effect of Re-init is not local; even re-initializing only the topmost Transformer block can affect the whole network. While setting $L = 1$ or $L = 3$ continues to benefit the bottom Transformer blocks, re-initializing too many layers (e.g., $L = 10$) can increase the $L2$ distance in the bottom Transformer blocks, suggesting a tradeoff between the bottom and the top Transformer blocks. Collectively, these results suggest that Re-init finds a better initialization for fine-tuning and the top $L$ layers of BERT are potentially overspecialized to the pre-training objective.

## 6 TRAINING ITERATIONS: FINE-TUNING BERT FOR LONGER

BERT is typically fine-tuned with a slanted triangular learning rate, which applies linear warm-up to the learning rate followed by a linear decay. This learning schedule warrants deciding the number of training iterations upfront. Devlin et al. (2019) recommend fine-tuning GLUE datasets for three epochs. This recommendation has been adopted broadly for fine-tuning (Phang et al., 2018; Lee et al., 2020; Dodge et al., 2020). We study the impact of this choice, and observe that this one-size-fits-all

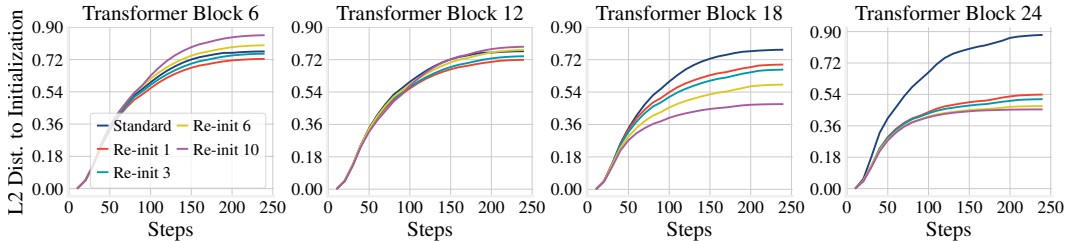

Figure 7: $L2$ distance to the initial parameters during fine-tuning BERT on RTE. Re-init reduces the amount of change in the weights of top Transformer blocks. However, re-initializing too many layers causes a larger change in the bottom Transformer blocks.

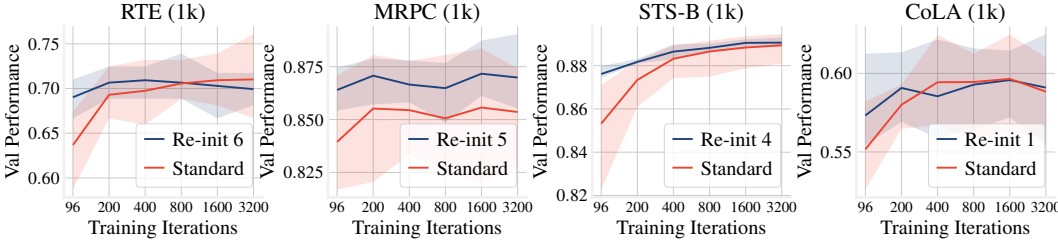

Figure 8: Mean (solid lines) and range (shaded region) of validation performance trained with different number of iterations, across eight random trials.

three-epochs practice for BERT fine-tuning is sub-optimal. Fine-tuning BERT longer can improve both training stability and model performance.

**Experimental setup** We study the effect of increasing the number of fine-tuning iterations for the datasets in Section 3. For the 1k downsampled datasets, where three epochs correspond to 96 steps, we tune the number of iterations in $\{200, 400, 800, 1600, 3200\}$. For the four small datasets, we tune the number of iterations in the same range but skip values smaller than the number of iterations used in three epochs. We evaluate our models ten times on the validation set during fine-tuning. This number is identical to the experiments in Sections 4–5, and controls for the set of models to choose from. We tune with eight different random seeds and select the best set of hyperparameters based on the mean validation performance to save experimental costs. After the hyperparameter search, we fine-tune with the best hyperparameters for 20 seeds and report the test performance.

**Results** Table 1 shows the result under the *Longer* column. Training longer can improve over the three-epochs setup most of the time, in terms of both performance and stability. This is more pronounced on the 1k downsampled datasets. We also find that training longer reduces the gap between standard fine-tuning and Re-init, indicating that training for more iterations can help these models recover from bad initializations. However, on datasets such as MRPC and MNLI, Re-init still improves the final performance even with training longer. We show the validation results on the four downsampled datasets with different number of training iterations in Figure 8. We provide a similar plot in Figure 14, Appendix G for the other downsampled datasets. We observe that different tasks generally require different number of training iterations and it is difficult to identify a one-size-fits-all solution. Therefore, we recommend practitioners to tune the number of training iterations on their datasets when they discover instability in fine-tuning. We also observe that on most of the datasets, Re-init requires fewer iterations to achieve the best performance, corroborating that Re-init provides a better initialization for fine-tuning.

## 7 REVISITING EXISTING METHODS FOR FEW-SAMPLE BERT FINE-TUNING

Instability in BERT fine-tuning, especially in few-sample settings, is receiving increasing attention recently (Devlin et al., 2019; Phang et al., 2018; Lee et al., 2020; Dodge et al., 2020). We revisit these methods given our analysis of the fine-tuning process, focusing on the impact of using the debiased ADAM instead of BERTADAM (Section 4). Generally, we find that when these methods are re-evaluated with the unbiased ADAM they are less effective with respect to the improvement in fine-tuning stability and performance.

|  | Standard | Int. Task | LLRD | Mixout | Pre-trained WD | WD | Re-init | Longer |
|---|---|---|---|---|---|---|---|---|
| RTE | $69.5 \pm 2.5$ | $81.8 \pm 1.7$ | $69.7 \pm 3.2$ | $71.3 \pm 1.4$ | $69.6 \pm 2.1$ | $69.5 \pm 2.5$ | $72.6 \pm 1.6$ | $72.3 \pm 1.9$ |
| MRPC | $90.8 \pm 1.3$ | $91.8 \pm 1.0$ | $91.3 \pm 1.1$ | $90.4 \pm 1.4$ | $90.8 \pm 1.3$ | $90.8 \pm 1.3$ | $91.4 \pm 0.8$ | $91.0 \pm 1.3$ |
| STS-B | $89.0 \pm 0.6$ | $89.2 \pm 0.3$ | $89.2 \pm 0.4$ | $89.2 \pm 0.4$ | $89.0 \pm 0.5$ | $89.0 \pm 0.6$ | $89.4 \pm 0.2$ | $89.6 \pm 0.3$ |
| CoLA | $63.0 \pm 1.5$ | $63.9 \pm 1.8$ | $63.0 \pm 2.5$ | $61.6 \pm 1.7$ | $63.4 \pm 1.5$ | $63.0 \pm 1.5$ | $64.2 \pm 1.6$ | $62.4 \pm 1.7$ |

Table 2: Mean test performance and standard deviation on four datasets. Numbers that are statistically significantly better than the standard setting (left column) are in blue and underlined. The results of Re-init and Longer are copied from Table 1. All experiments use ADAM with debiasing (Section 4). Except Longer, all methods are trained with three epochs. "Int. Task" stands for transfering via an intermediate task (MNLI).

## 7.1 Overview

**Pre-trained Weight Decay** Weight decay (WD) is a common regularization technique (Krogh & Hertz, 1992). At each optimization iteration, $\lambda \mathbf{w}$ is subtracted from the model parameters, where $\lambda$ is a hyperparameter for the regularization strength and $\mathbf{w}$ is the model parameters. Pre-trained weight decay adapts this method for fine-tuning pre-trained models (Chelba & Acero, 2004; Daumé III, 2007) by subtracting $\lambda(\mathbf{w} - \hat{\mathbf{w}})$ from the objective, where $\hat{\mathbf{w}}$ is the pre-trained parameters. Lee et al. (2020) empirically show that pre-trained weight decay works better than conventional weight decay in BERT fine-tuning and can stabilize fine-tuning.

**Mixout** Mixout (Lee et al., 2020) is a stochastic regularization technique motivated by Dropout (Srivastava et al., 2014) and DropConnect (Wan et al., 2013). At each training iteration, each model parameter is replaced with its pre-trained value with probability $p$. The goal is to prevent catastrophic forgetting, and (Lee et al., 2020) proves it constrains the fine-tuned model from deviating too much from the pre-trained initialization.

**Layer-wise Learning Rate Decay (LLRD)** LLRD (Howard & Ruder, 2018) is a method that applies higher learning rates for top layers and lower learning rates for bottom layers. This is accomplished by setting the learning rate of the top layer and using a multiplicative decay rate to decrease the learning rate layer-by-layer from top to bottom. The goal is to modify the lower layers that encode more general information less than the top layers that are more specific to the pre-training task. This method is adopted in fine-tuning several recent pre-trained models, including XLNet (Yang et al., 2019b) and ELECTRA (Clark et al., 2020).

**Transferring via an Intermediate Task** Phang et al. (2018) propose to conduct supplementary fine-tuning on a larger, intermediate task before fine-tuning on few-sample datasets. They show that this approach can reduce variance across different random trials and improve model performance. Their results show that transferring models fine-tuned on MNLI (Williams et al., 2018) can lead to significant improvement on several downstream tasks including RTE, MRPC, and STS-B. In contrast to the other methods, this approach requires large amount of additional annotated data.

## 7.2 Experiments

We evaluate all methods on RTE, MRPC, STS-B, and CoLA. We fine-tune a BERT$_{\text{Large}}$ model using the ADAM optimizer with debiasing for three epochs, the default number of epochs used with each of the methods. For intermediate task fine-tuning, we fine-tune a BERT$_{\text{Large}}$ model on MNLI and then fine-tune for our evaluation. For other methods, we perform hyperparameter search with a similar size search space for each method, as described in Appendix H. We do model selection using the average validation performance across 20 random seeds. We additionally report results for standard fine-tuning with longer training time (Section 6), weight decay, and Re-init (Section 5).

Table 2 provides our results. Compared to published results (Phang et al., 2018; Lee et al., 2020), our test performance for Int. Task (transferring via an intermediate task), Mixout, Pre-trained WD, and WD are generally higher when using the ADAM with debiasing.[10] However, we observe less pronounced benefits for all surveyed methods compared to results originally reported. At

---

[10]The numbers in Table 2 are not directly comparable with previously published validation results (Phang et al., 2018; Lee et al., 2020) because we are reporting test performance. However, the relatively large margin between our results and previously published results indicates an improvement. More important, our focus is the relative improvement, or lack of improvement compared to simply training longer.

times, these methods do not outperform the standard baselines or simply training longer. Using additional annotated data for intermediate task training continues to be effective, leading to consistent improvement over the average performance across all datasets. LLRD and Mixout show less consistent performance impact. We observe no noticeable improvement using pre-trained weight decay and conventional weight decay in improving or stabilizing BERT fine-tuning in our experiments, contrary to existing work (Lee et al., 2020). This indicates that these methods potentially ease the optimization difficulty brought by the debiasing omission in BERTADAM, and when we add the debiasing, the positive effects are reduced.

## 8 CONCLUSION

We have demonstrated that optimization plays a vital role in the few-sample BERT fine-tuning. First, we show that the debiasing omission in BERTADAM is the main cause of degenerate models on small datasets commonly observed in previous work (Phang et al., 2018; Lee et al., 2020; Dodge et al., 2020). Second, we observe the top layers of the pre-trained BERT provide a detrimental initialization for fine-tuning and delay learning. Simply re-initializing these layers not only speeds up learning but also leads to better model performance. Third, we demonstrate that the common one-size-fits-all three-epochs practice for BERT fine-tuning is sub-optimal and allocating more training time can stabilize fine-tuning. Finally, we revisit several methods proposed for stabilizing BERT fine-tuning and observe that their positive effects are reduced with the debiased ADAM. In the future, we plan to extend our study to different pre-training objectives and model architectures, and study how model parameters evolve during fine-tuning.

ACKNOWLEDGMENTS

We thank Cheolhyoung Lee for his help in reproducing previous work. We thank Lili Yu, Ethan R. Elenberg, Varsha Kishore, and Rishi Bommasani for their insightful comments, and Hugging Face for the Transformers project, which enabled our work.

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

| Task | RTE NLI | MRPC Paraphrase | STS-B Similarity | CoLA Acceptibility | SST-2 Sentiment | QNLI NLI | QQP Paraphrase | MNLI NLI |
|---|---|---|---|---|---|---|---|---|
| # of training samples | 2.5k | 3.7k | 5.8k | 8.6k | 61.3k | 104k | 363k | 392k |
| # of validation samples | 139 | 204 | 690 | 521 | 1k | 1k | 1k | 1k |
| # of test samples | 139 | 205 | 690 | 521 | 1.8k | 5.5k | 40k | 9.8k |
| Evaluation metric | Acc. | F1 | SCC | MCC | Acc. | Acc. | Acc. | Acc. |
| Majority baseline (val) | 52.9 | 81.3 | 0 | 0 | 50.0 | 50.0 | 50.0 | 33.3 |
| Majority baseline (test) | 52.5 | 81.2 | 0 | 0 | 49.1 | 50.5 | 63.2 | 31.8 |

Table 3: The datasets used in this work. We apply non-standard data splits to create test sets. SCC stands for Spearman Correlation Coefficient and MCC stands for Matthews Correlation Coefficient.

## A  DATASETS

Table 3 summarizes dataset statistics and describes our validation/test splits. We also provide a brief introduction for each datasets:

**RTE**  Recognizing Textual Entailment (Bentivogli et al., 2009) is a binary entailment classification task. We use the GLUE version.

**MRPC**  Microsoft Research Paraphrase Corpus (Dolan & Brockett, 2005) is binary classification task. Given a pair of sentences, a model has to predict whether they are paraphrases of each other. We use the GLUE version.

**STS-B**  Semantic Textual Similarity Benchmark (Cer et al., 2017) is a regression tasks for estimating sentence similarity between a pair of sentences. We use the GLUE version.

**CoLA**  Corpus of Linguistic Acceptability (Warstadt et al., 2019) is a binary classification task for verifying whether a sequence of words is a grammatically correct English sentence. Matthews correlation coefficient (Matthews, 1975) is used to evaluate the performance. We use the GLUE version.

**MNLI**  Multi-Genre Natural Language Inference Corpus (Williams et al., 2018) is a textual entailment dataset, where a model is asked to predict whether the premise entails the hypothesis, predicts the hypothesis, or neither. We use the GLUE version.

**QQP**  Quora Question Pairs (Iyer et al., 2017) is a binary classification task to determine whether two questions are semantically equivalent (i.e., paraphrase each other). We use the GLUE version.

**SST-2**  The binary version of the Stanford Sentiment Treebank (Socher et al., 2013) is a binary classification task for whether a sentence has positive or negative sentiment. We use the GLUE version.

## B  ISOLATING THE IMPACT OF DIFFERENT SOURCES OF RANDOMNESS

The randomness in BERT fine-tuning comes from three sources: (a) weight initialization, (b) data order, and (c) Dropout regularization (Srivastava et al., 2014). We control the randomness using two separate random number generators: one for weight initialization and the other for both data order and Dropout (both of them affect the stochastic loss at each iteration). We fine-tune BERT on RTE for three epochs using ADAM with 10 seeds for both random number generators. We compare the standard setup with Re-init 5, where $L = 5$. This experiment is similar to Dodge et al. (2020), but we use ADAM with debiasing instead of BERTADAM and control for the randomness in Dropout as well. When fixing a random seed for weight initialization, Re-init 5 shares the same initialized classifier weights with the standard baseline. Figure 9 shows the validation accuracy of each individual run as well as the minimum, average, and maximum scores when fixing one of the random seeds. Figure 10 summarizes the standard deviations when one of the random seeds is controlled. We observe several trends. Re-init 5 usually improves the performance regardless of the weight initialization or data order and Dropout. Second, Re-init 5 still reduces the instability when one of the sources of randomness is controlled. Third, the standard deviation of fixing the weight initialization roughly matches the one of controlled data order and Dropout, which aligns with the observation of Dodge et al. (2020).

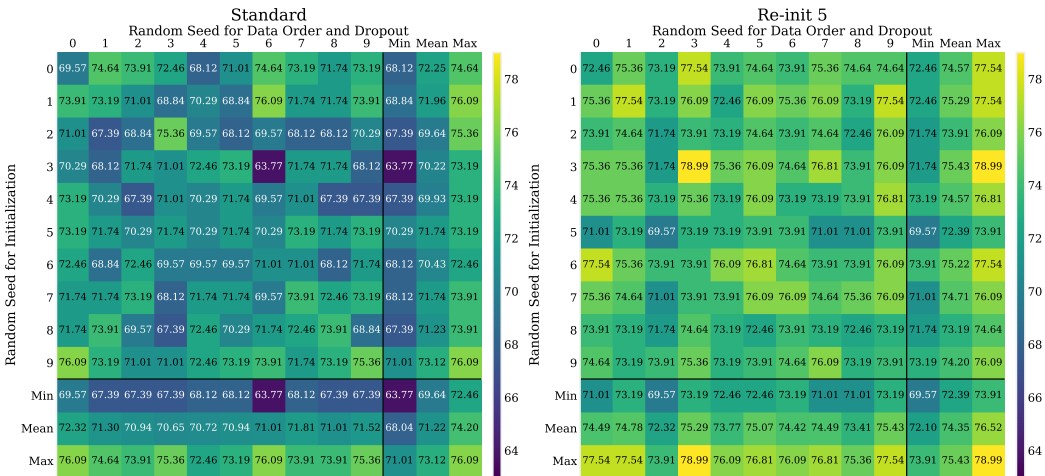

Figure 9: Validation accuracy on RTE with controlled random seeds. The min, mean, and max values of controlling one of the random seeds are also included. Re-init 5 usually improves the validation accuracy.

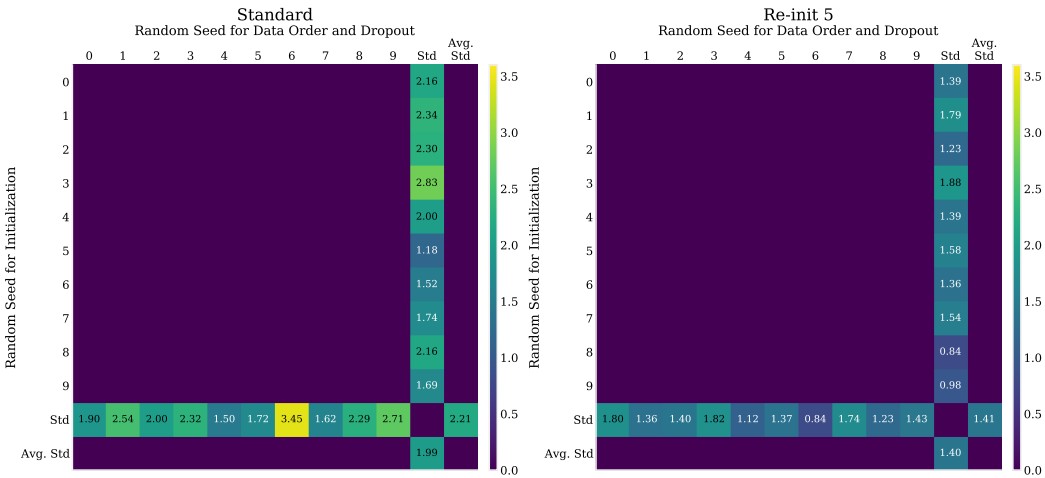

Figure 10: The standard deviation of the validation accuracy on RTE with controlled random seeds. We show the standard deviation of fixing either the initialization or data order and Dropout. Re-init 5 consistently reduces the instability regardless of the sources of the randomness.

## C    MIXED PRECISION TRAINING

Mixed precision training can accelerate model training while preserving performance by replacing some 32-bit floating-point computation with 16-bit floating-point computation. We use mixed precision training in all our experiments using huggingface's Transformers (Wolf et al., 2019). Transformers uses O1-level optimized mixed precision training implemented with the Apex library.[11] We evaluate if this mixed precision implementation influences our results. We fine-tune BERT with 20 random trials on RTE, MRPC, STS-B, and CoLA. We use two-tailed $t$-test to test if the distributions of the two methods are statistically different. Table 4 shows the mean and standard deviation of the

---

[11]https://github.com/NVIDIA/apex

|  | CoLA | MRPC | RTE | STS-B |
|---|---|---|---|---|
| Mixed precision | $60.3 \pm 1.5$ | $89.2 \pm 1.2$ | $71.8 \pm 2.1$ | $90.1 \pm 0.7$ |
| Full precision | $59.9 \pm 1.5$ | $88.7 \pm 1.4$ | $71.4 \pm 2.2$ | $90.1 \pm 0.7$ |

Table 4: Comparing BERT fine-tuning with mixed precision and full precision. The difference between the two numbers on any dataset is not statistically significant.

|  | Dev Acc. (%) | Test Acc. (%) |
|---|---|---|
| No bias correction | $86.0 \pm 0.3$ | $87.0 \pm 0.4$ |
| Bias correction | $85.9 \pm 0.3$ | $86.9 \pm 0.3$ |

Table 5: Comparing BERT fine-tuning with and without bias correction on the MNLI dataset. When we have a large dataset, there is no significant difference in using bias correction or not.

test performance. The performance of mixed precision matches the single precision counterpart, and there is no statistically significant difference.

## D    BIAS-CORRECTION ON MNLI

The focus of this paper is few-sample learning. However, we also experiment with the full MNLI dataset. Table 5 shows that average accuracy over three random runs. The results confirm that there is no significant difference in using bias correction or not on such a large dataset. While our recommended practices do not improve training on large datasets, this result shows there is no disadvantage to fine-tune such models with the same procedure as we propose for few-sample training.

## E    SUPPLEMENTARY MATERIAL FOR SECTION 4

**Effect of ADAM with Debiasing on Convergence.**    Figure 11 shows the training loss as a function of the number of training iterations. Using bias correction effectively speeds up convergence and reduces the range of the training loss, which is consistent with our observation in Figure 3.

**Effect of ADAM with Debiasing on the Expected Validation Performance.**    Figure 12 shows the expected validation performance as a function of the number of random trials. Comparing to Figure 4, we observe several trends. First, using ADAM with debiasing consistently leads to faster convergence and improved validation performance, which is similar to our observation about the test performance. Second, we observe that the expected validation performance monotonically increases with the number of random trials, contrary to our observation about the test performance. This suggests that using too many random trials may overfit to the validation set and hurt generalization performance.

## F    SUPPLEMENTARY MATERIAL FOR SECTION 5

**Effect of $L$ on Re-init**    Figure 13 shows the effect of Re-init in fine-tuning on the eight downsampled datasets. We observe similar trends in Figure 13 and Figure 5. Re-init's improvement is more pronounced in the wort-case performance across different random trials. Second, the best value of $L$ is different for each dataset.

**Effect of Re-init on Model Parameters**    We use the same setup as in Figure 7 to plot the change in the weights of different Transformer blocks during fine-tuning on RTE, MRPC, STS-B, and CoLA in Figures 15–18.

**Effect of Re-init on Other Models**    We study more recent pre-trained contexual embedding models beyond $BERT_{Large}$. We investigate whether Re-init provides better fine-tuning initialization in $XLNet_{Large}$ Yang et al. (2019b), $RoBERTa_{Large}$ Liu et al. (2019c), $BART_{Large}$ Lewis et al. (2019), and $ELECTRA_{Large}$ Clark et al. (2020). XLNet is an autoregressive language model trained by learning

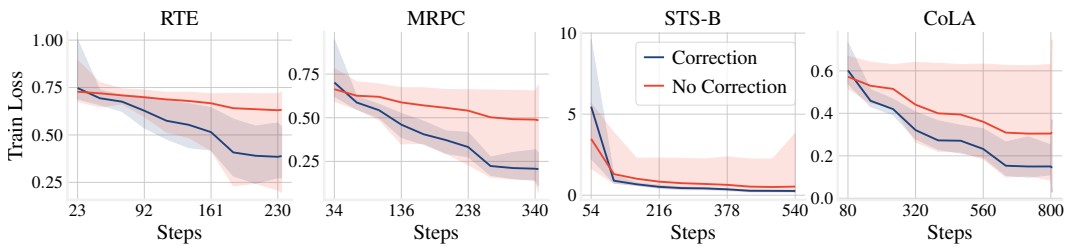

Figure 11: Mean (solid lines) and range (shaded region) of training loss during fine-tuning BERT, across 50 random trials. Bias correction speeds up convergence and reduces the range of the training loss.

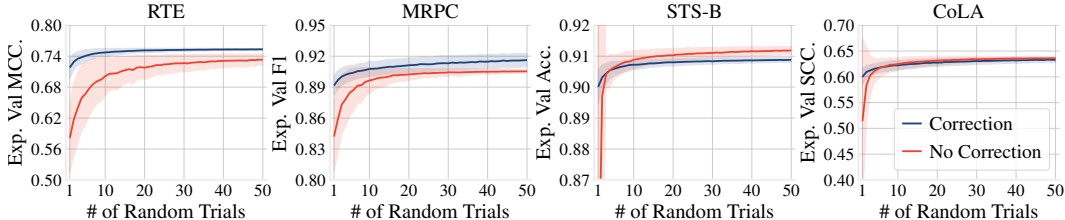

Figure 12: Expected validation performance (solid lines) with standard deviation (shaded region) over the number of random trials allocated for fine-tuning BERT. With bias correction, we can reliably achieve good results with few (i.e., 5 or 10) random trials.

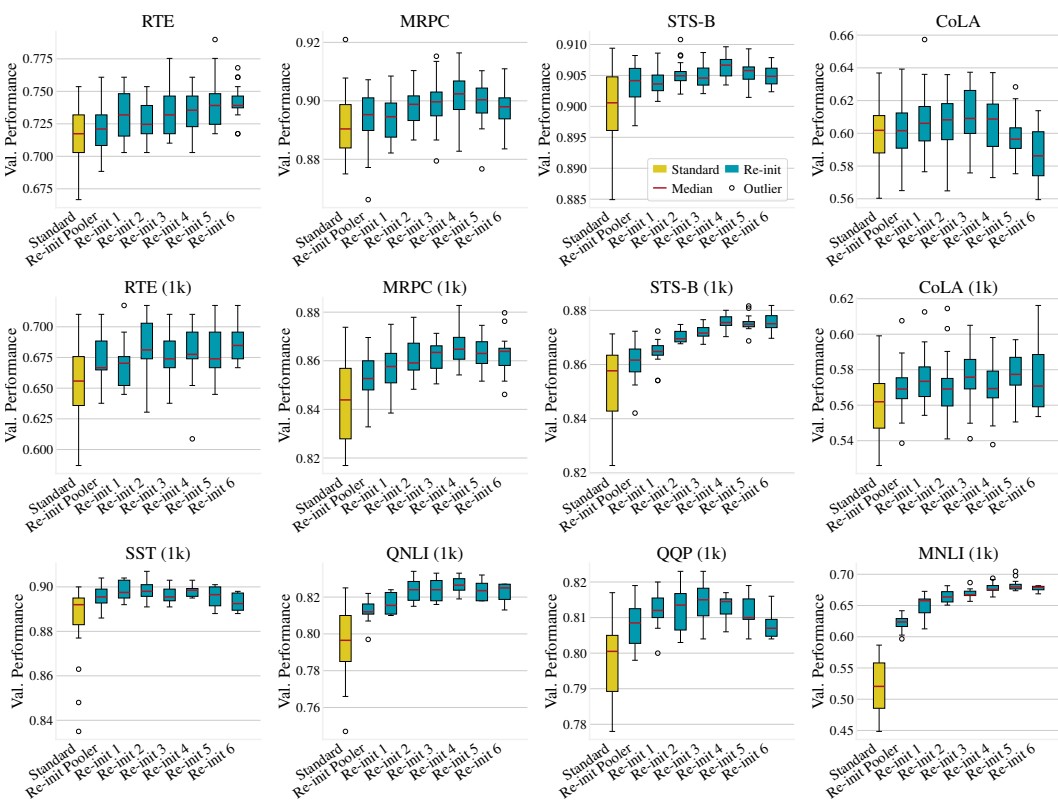

Figure 13: Validation performance distribution of re-initializing different number of layers of BERT on the downsampled datasets.

all permutations of natural language sentences. RoBERTa is similar to BERT in terms of model architecture but is only pre-trained on the mask language modeling task only, but for longer and on

| Model | Dataset | Learning Rate | Training Epochs / Steps | Batch Size | Warmup Ratio / Steps | LLRD |
|---|---|---|---|---|---|---|
| BERT | all | $2 \times 10^{-5}$ | 3 epochs | 32 | 10% | - |
| XLNet | RTE | $3 \times 10^{-5}$ | 800 steps | 32 | 200 steps | - |
|  | MRPC | $5 \times 10^{-5}$ | 800 steps | 32 | 200 steps | - |
|  | STS-B | $5 \times 10^{-5}$ | 3000 steps | 32 | 500 steps | - |
|  | CoLA | $3 \times 10^{-5}$ | 1200 steps | 128 | 120 steps | - |
| RoBERTa | RTE | $2 \times 10^{-5}$ | 2036 steps | 16 | 122 steps | - |
|  | MRPC | $1 \times 10^{-5}$ | 2296 steps | 16 | 137 steps | - |
|  | STS-B | $2 \times 10^{-5}$ | 3598 steps | 16 | 214 steps | - |
|  | CoLA | $1 \times 10^{-5}$ | 5336 steps | 16 | 320 steps | - |
| ELECTRA | RTE | $5 \times 10^{-5}$ | 10 epochs | 32 | 10% | 0.9 |
|  | MRPC | $5 \times 10^{-5}$ | 3 epochs | 32 | 10% | 0.9 |
|  | STS-B | $5 \times 10^{-5}$ | 10 epochs | 32 | 10% | 0.9 |
|  | CoLA | $5 \times 10^{-5}$ | 3 epochs | 32 | 10% | 0.9 |
| BART | RTE | $1 \times 10^{-5}$ | 1018 steps | 32 | 61 steps | - |
|  | MRPC | $2 \times 10^{-5}$ | 1148 steps | 64 | 68 steps | - |
|  | STS-B | $2 \times 10^{-5}$ | 1799 steps | 32 | 107 steps | - |
|  | CoLA | $2 \times 10^{-5}$ | 1334 steps | 64 | 80 steps | - |

Table 6: Fine-tuning hyper-parameters of BERT and its variants as reported in the official repository of each model.

| | RTE | | MRPC | | STS-B | | CoLA | |
|---|---|---|---|---|---|---|---|---|
| | Standard | Re-init | Standard | Re-init | Standard | Re-init | Standard | Re-init |
| XLNet | $71.7 \pm 12.6$ | $80.1 \pm 1.6$ | $92.3 \pm 4.3$ | $94.5 \pm 0.8$ | $86.8 \pm 20.4$ | $91.7 \pm 0.3$ | $51.8 \pm 22.5$ | $62.0 \pm 2.1$ |
| RoBERTa | $78.2 \pm 12.1$ | $83.5 \pm 1.4$ | $94.4 \pm 0.9$ | $94.8 \pm 0.9$ | $91.8 \pm 0.3$ | $91.8 \pm 0.2$ | $68.4 \pm 2.2$ | $67.6 \pm 1.5$ |
| ELECTRA | $87.1 \pm 1.2$ | $86.1 \pm 1.9$ | $95.7 \pm 0.8$ | $95.3 \pm 0.8$ | $91.8 \pm 1.9$ | $92.1 \pm 0.5$ | $62.1 \pm 20.4$ | $61.3 \pm 20.1$ |
| BART | $84.1 \pm 2.0$ | $83.5 \pm 1.5$ | $93.5 \pm 0.9$ | $93.7 \pm 1.2$ | $91.7 \pm 0.3$ | $91.8 \pm 0.3$ | $65.4 \pm 1.9$ | $64.9 \pm 2.3$ |

Table 7: Average Test performance with standard deviation on four small datasets with four different pre-trained models. For each setting, the better numbers are bolded and are in blue if the improvement is statistically significant.

more data. BART is a sequence-to-sequence model trained as a denoising autoencoder. ELECTRA is a BERT-like model trained to distinguish tokens generated by masked language model from tokens drawn from the natural distribution. Together, they represent a diverse range of modeling choices in pre-training, including different model architectures, objectives, data, and training strategies. We use ADAM with debiasing to fine-tune these models on RTE, MRPC, STS-B, and COLA, using the hyperparameters that are either described in the paper or in the official repository of each model. Table 6 summarizes the hyper-parameters of each model for each dataset. We use the huggingface's Transformers library (Wolf et al., 2019). The experimental setup is kept the same as our other experiments. Table 7 displays the average test performance on these datasets. We observe that several models suffer from high instability on these datasets and in most cases, Re-init can reduce the performance variance. We observe that for some models, like XLNet$_{\text{Large}}$ or RoBERTa$_{\text{Large}}$, Re-init can improve the average performance and reduce variance. However, the behavior of Re-init varies significantly across different models and Re-init have less significant improvement for ELECTRA$_{\text{Large}}$ and BART$_{\text{Large}}$. Further study of the entire model family requires significant computational resources, and we leave it as an important direction for future work.

# G SUPPLEMENTARY MATERIAL FOR SECTION 6

Figure 14 plots the validation performance as a function of the number of training iteration, using the same setting as in Figure 8. Similar to our observations in Figure 8, we find that training longer generally improves fine-tuning performance and reduces the gap between standard fine-tuning and Re-init. On MNLI, Re-init still outperforms standard fine-tuning.

Figure 14: Mean (solid lines) and range (shaded region) of validation performance trained with different number of iterations, across eight random trials.

# H    EXPERIMENTAL DETAILS IN SECTION 7

The hyperparameter search space allocated for each method in our experiments is:

**Layerwise Learning Rate Decay (LLRD)**    We grid search the initial learning rate in $\{2\times10^{-5}, 5\times 10^{-5}, 1 \times 10^{-4}\}$ and the layerwise decay rate in $\{0.9, 0.95\}$.

**Mixout**    We tune the mixout probability $p \in \{0.1, 0.3, 0.5, 0.7, 0.9\}$.

**Weight decay toward the pre-trained weight**    We tune the regularization strength $\lambda \in \{10^{-3}, 10^{-2}, 10^{-1}, 10^0\}$.

**Weight decay**    We tune the regularization strength $\lambda \in \{10^{-4}, 10^{-3}, 10^{-2}, 10^{-1}\}$.

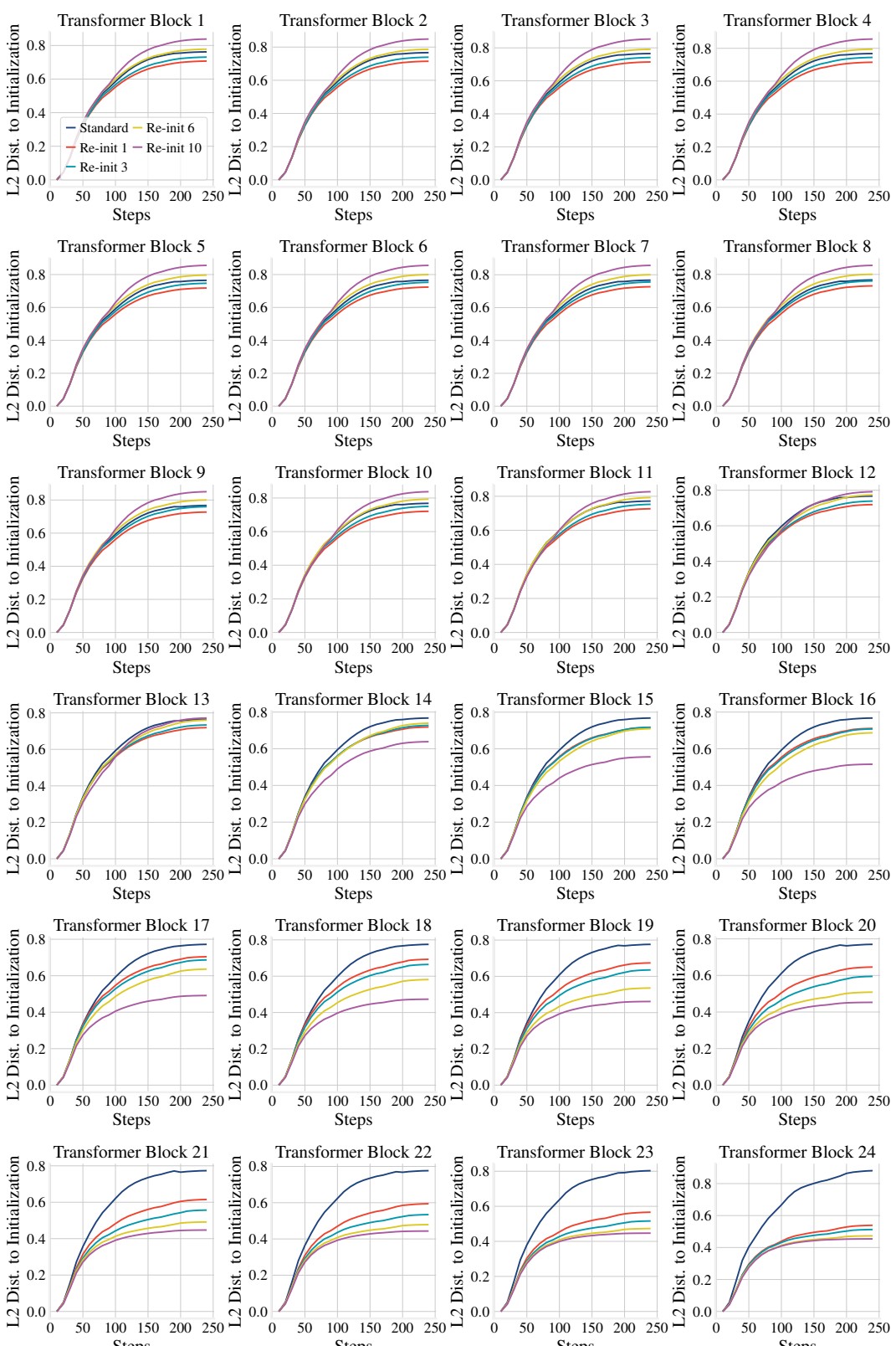

Figure 15: $L2$ distance to the initialization during fine-tuning BERT on RTE.

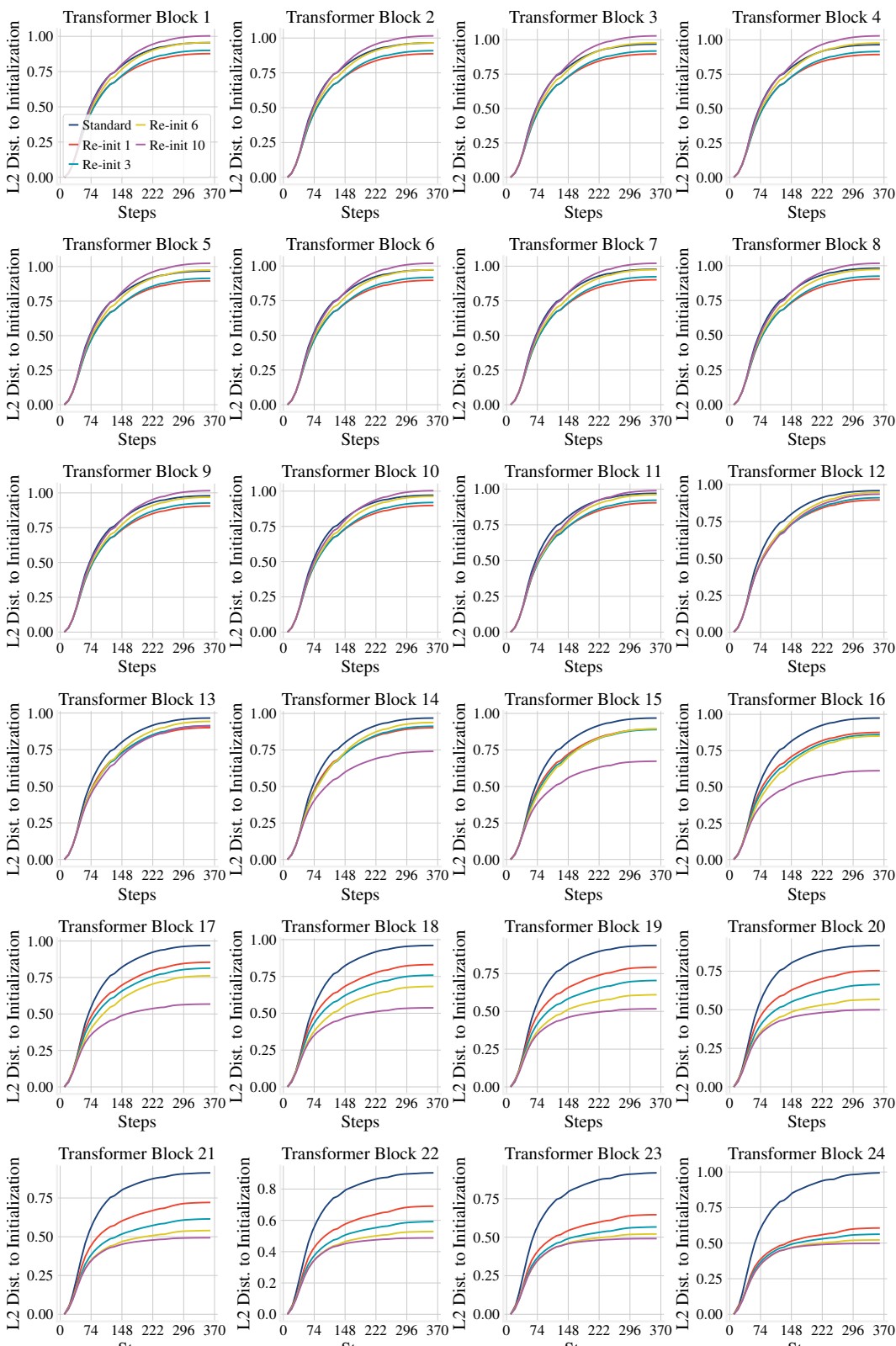

Figure 16: $L2$ distance to the initialization during fine-tuning BERT on MRPC.

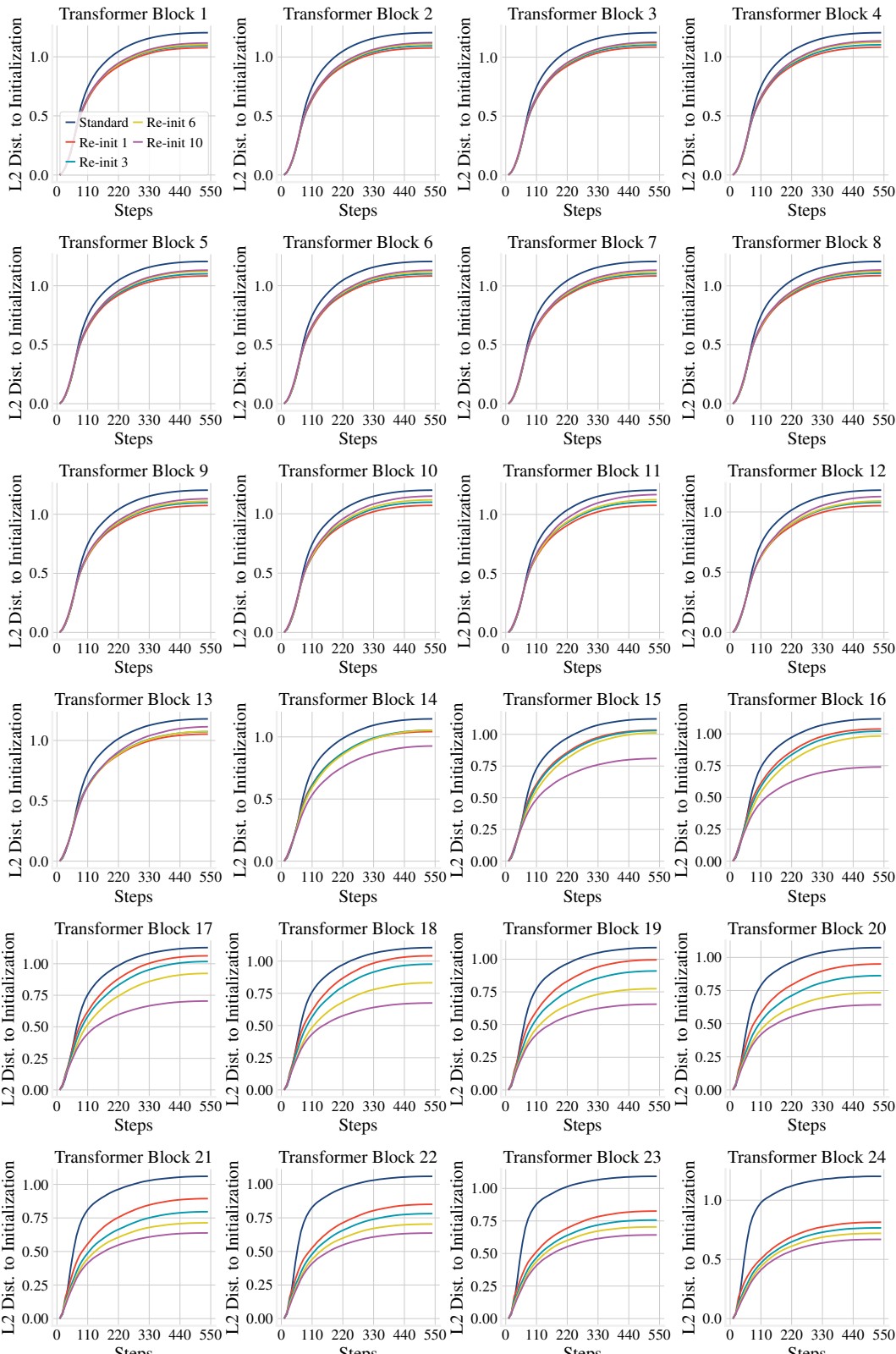

Figure 17: $L2$ distance to the initialization during fine-tuning BERT on STS-B.

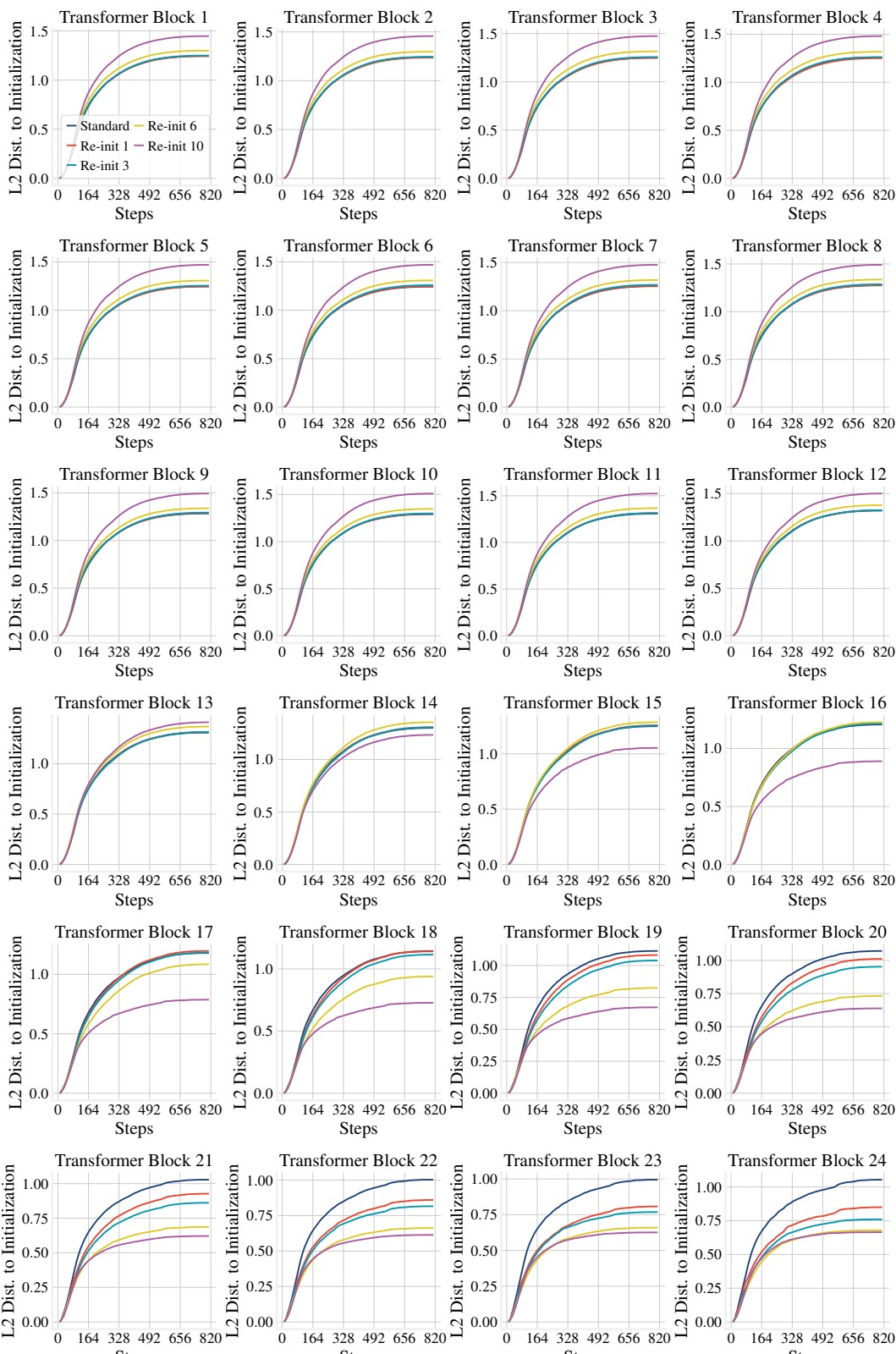

Figure 18: $L2$ distance to the initialization during fine-tuning BERT on CoLA.

