# OpenReview forum: "Revisiting Few-sample BERT Fine-tuning"
_ICLR.cc/2021/Conference — ICLR 2021 Poster_

### Official Review · AnonReviewer4 · 2020-10-27
**An exhaustive study on the instabilities of BERT fine-tuning, and simple and intuitive methods to circumvent these problems and improve the average performance of fine-tuning on different tasks**

**Rating:** 7
**Confidence:** 3

**Review:**

Large language models (LM) architectures, such as BERT, XLNet, etc., are not generally trained from scratch, but rather used as pretrained models. Among all, BERT is one of the most widely used ones, and its use on downstream tasks mainly consists on a stage of fine-tuning, where the new layers added are trained, and the rest of parameters of the network are left unfrozen, and hence are adjusted slightly to better fit the new task. However, this step of fine-tuning BERT is known to be quite unstable, and depends on a large set of factors, especially the initialization. Since the final performance on these downstream tasks can vary notably, different approaches has been proposed to circumvent this, but still the most common solution consists simply on choosing the best performing model, from a few random initialisation, using the validation set.

##### Summary
In the current paper, the authors aim at tackling the aforementioned problem in a more grounded way. They investigate deeper the possible causes for these instabilities, and propose methods to counteract these pitfalls. Hence, they propose three simple, yet effective, approaches to stabilise the training and ensure better performances: a modified optimiser, the use of randomly initialised top layers, and more training steps. They provide a large collection of results, compare all these solutions to previous works, and discuss differences and similarities. Thanks to the analyses carried out, the current paper results in an exhaustive study on how “safely” fine-tune BERT, and the different factors that are to be taken into account when making use of these models.

##### Strong and weak points
I would like to start with the weakest point of the paper: it actually does not present anything clearly novel, nor innovative or groundbreaking. All the solutions proposed are inspired by previous approaches, or are just slight modifications of existing methods. But, this does not mean the paper is not valuable, as I do believe it is. The instability while fine-tuning large LMs on downstream tasks is a well known problem, but yet it has not been tackle exhaustively, and I do believe there does not exist clear guidelines and/or modifications that enable easily circumventing a critical weakness of these models. But I consider this paper succeeds at precisely this important task, thanks to the extended and exhaustive study it presents, and how it proposes three simple modifications that seem to solve this pitfall on most scenarios.

Besides, the paper is quite well written, and presents in a clear manner the problems with the models, some intuition about the cause of those issues, and then, the solutions to overcome them. All the solutions are sufficiently justified, and are intuitive and simple. The latter, instead of being a weak point, for this precise problem it is more an advantage, as will allow an effortless adoption. Its improved performance is ensured thanks to the large set of benchmarks, on various datasets, the authors have compiled on the current manuscript. This is indeed another strong point, as all the solutions proposed are also tested under different conditions, with more or less training steps, and different numbers of top layers randomly initialised.

##### Decision, and key reasons
I believe the paper is ready to be accepted. Overall, it is an interesting and useful paper that will help many NLP researchers, and end-users of BERT, fine-tune better models, obtain improved performances, and therefore, start from a better baseline for their endeavours. And all this, with just some simple and intuitive modifications and guidelines. All the proposed methods and suggestions are not drawn from a few bunch of tests, but rather from a large collection of simulations, for different and varied datasets, with disparate starting conditions, and run over a fair amount of random initialisations. Therefore, I believe the authors have taken their time, and simulation time, to ensure that the presented results are robust and consistent, which is something to remark also.

##### Questions and additional evidence
Although I believe the paper is nicely written, and compiles all the required results and tests, I would appreciate if the authors could comment further on the following points:
* I do believe there is a reason for not performing bias-correction on BERTAdam, and therefore, introducing it back might be affecting BERT training and fine-tuning in some specific, I guess negative, way. Could the authors comment on this? Or their understanding on why the correction was removed for BERTAdam.
* In Figure 4, you suggest that with 5 to 10 random trials, the bias correction will achieve good results. However, observing the plots for all the datasets, we realise that indeed that number of random trials may benefit more the non-corrected version, as in most of the datasets the performance is either higher, or at least comparable. And although the variance is larger, we might still ensure at least a similar result . Could you comment on this? Would not be the corrected version a better option when no random initialisations are envisaged?
* For the re-init, when just training for 3 epochs, it surprises me that indeed we could train the last 6 layers with just this reduced amount of data and training steps. And more surprisingly, according to Figures 14-16, is that the weights for these last 6 layers are the first to stabilise, even though they started from scratch, and they are supposed to be critical for the downstream tasks. Could you comment on this? I guess my understanding is wrong, and I would appreciate therefore some further insights.
* Also, on the Re-init scheme, you mention that the number of layers to re-initialize depends on the task. Could you in any case offer here a general rule of thumb?

##### Extra feedback
Finally, I would like to conclude listing some small typos and errors I could spot in the manuscript:
* Page 7, after Results, the reference to the Table is wrong.
* Page 8, table 2: I believe the result for the RTE - Int. Task is mistype. I guess it should be something around 71.8.
* Page 14, section E, Effect of Re-init… : the reference to the figure.
* The caption for all figures 14 to 17 is wrong, as it should read fine-tuning.

These are the ones I could find, but it is not an exhaustive list. In any case, I would like to highlight the quality of the present manuscript, in terms of clearness and writing.

---

> ### Author Response · Authors · 2020-11-17
> **Response to Reviewer 4**
>
> Thank you for your detailed comments.
>
> Unfortunately, we do not have a definitive answer for why the bias correction was removed in the first place. Historically, this change is introduced in the Google BERT repo without any comment/explanation (https://github.com/google-research/bert/blob/master/optimization.py).
>
> As suggested, we carried out experiments on larger GLUE datasets to investigate the effect of bias correction (please see general reply). These results validate that bias correction does not have a negative impact for fine-tuning larger datasets.
>
> Given enough random trials, the *best* model obtained by uncorrected Adam is comparable to the *best* model obtained with bias correction. This takes more experiments though, which carry costs. Given that there is minimal cost to applying bias correction, we recommend using it as a standard practice.
>
> Your understanding is correct. While we do not have a clear answer for this phenomenon, we want to provide some intuition. One factor to consider is the usage of residual connections in transformers. The last six layers, although randomly initialized, only add residual changes to the representations generated from the lower layers. Intuitively then, the last six layers are required for less work when adjusting for the downstream task so a small number of fine-tuning updates already leads to good results.
>
> We observe that re-initializing the last 4 layers can typically achieve most of the benefit. However, in some cases, re-initializing the last 1 or 2 does better. Most importantly, counterintuitively, there are always several layers at the top that can be re-initialized.
>
> Thank you for the detailed feedback on typos and we have corrected them in the paper. Regarding the RTE - Int. Task, we have verified that the “81.8” is correct. The huge gain comes from the fact that RTE and MNLI are very similar tasks.

---

### Official Review · AnonReviewer3 · 2020-10-27

**Rating:** 6
**Confidence:** 4

**Review:**

### Summary
This paper investigates fine-tuning BERT for few-sample datasets. Notably, the authors find debiasing omission in BERT-adam. They find original debiased adam is better than BERT-adam. Besides, they also find re-initializing top layers can speed up learning and achieve better performance. These two findings are interesting. Another finding fine-tuning BERT for Longer is incremental to some extend.

### Strengths
 * The two findings mentioned above are notable.

* The authors conduct extensive experiments to support their claims.

### Weaknesses and Questions
* Table 1 shows the results of re-init but does not show re-init how many top layers for each task.
* I suggest the authors can investigate debiased adam and re-init on the datasets with enough samples, like MNLI or QNLI. If they can achieve slight improvement or at least do not degrade the performance, we can just conveniently use the same fine-tuning method for most datasets.
* Lack of explaining the meaning of Int. Task.

---

> ### Author Response · Authors · 2020-11-17
> **Response to Reviewer 3**
>
> Regarding Table 1:
> We do not provide the number of layers, since the best number of layers to re-init is task dependent. It would make the table harder to read. More detailed results of re-initializing different layers are provided in Figure 12.
>
> The “Int. Task” in Table 2 stands for “Transferring via an Intermediate Task”. Thank you for pointing out this ambiguity. We have updated the paper to clarify this.
>
> Regarding larger datasets: please see the general response.

---

### Official Review · AnonReviewer1 · 2020-10-27
**Thorough Analysis of Various Finetuning Factors**

**Rating:** 6
**Confidence:** 4

**Review:**

The paper focuses on instability issues in BERT finetuning on small datasets. They list three factors which leads to instability, and provide simple fixes for each:
1. Lack of bias correction term in BertAdam -- Fix was to use standard Adam
2. Using all pretrained layers for finetuning -- Reinitializing the last few layers before finetuning.
3. Training for a predefined number of epochs -- Train for a large number of epochs.

The fixes proposed reduces the variance in the results, and in most cases also improves performance. They also show that several proposed solutions to fix training instability lose their impact when the aforementioned fixes are incorporated.

Overall, I like the paper; the observation about reinitializing top layers of BERT was interesting and counter intuitive to me; and I think this will be the most important contribution of the paper.

Although not directly related to BERT, this paper (https://arxiv.org/pdf/1804.00247.pdf) also suggests training for longer epochs. This paper should be cited here. The tasks considered in the original BERT paper had large datasets, so I think the 2-3 epoch suggestion was tuned to those.

The result about BertAdam being unstable in low data settings, was a nice contribution. I feel this algorithm was also suggested considering the large datasets considered in the BERT paper.

---

> ### Author Response · Authors · 2020-11-17
> **Response to Reviewer 1**
>
> Thank you for the review and the additional reference. We have added the reference to the revised version.

---

### Official Review · AnonReviewer2 · 2020-10-28
**Useful tips of BERT fine-tuning for practitioners and insightful analysis**

**Rating:** 6
**Confidence:** 5

**Review:**

This paper proposes a few tricks to improve the stability of BERT fine-turning, which include a standard Adam optimizer (with bias correction), the top BERT layers re-initiation and longer training. It provides extensive study on the GLUE benchmark showing how important these tricks are for small tasks (such as RTE/MRPC) which have less 1K training samples. The paper is well written and provides an insightful analysis.

Although it provides several useful tips for practitioners, it lacks novelty: for example the adam bias correction is from the original adam paper (also pointed it out by [2]) and training longer helping the performance is also observed by [1]. Gradient clipping may also help stabilize the training and it will be great to have a discussion as well. At last, does these approaches help large tasks, such as MNLI/QQP? It will be great to have a few settings: experiments on small or large tasks.

[1] Nakkiran et al, Deep double descent: where bigger models and more data hurt.
[2] Mosbach et al, On the Stability of Fine-tuning BERT: Misconceptions, Explanations, and Strong Baselines

---

> ### Author Response · Authors · 2020-11-17
> **Response to Reviewer 2**
>
> Thank you for pointing out [1]. Although this paper studies training from scratch rather than fine-tuning, it is definitely related to our work. We have cited it in our revised version.
>
> Regarding the novelty concern:
> We are happy to see [2] has the same observation to ours, and the two papers validate each other. However, [2] is concurrent work submitted to this conference (https://openreview.net/forum?id=nzpLWnVAyah). In order not to omit their credits, we cite it in Section 2. Fortunately, their submitted version also cites us as concurrent work. We believe both papers are novel since they are developed concurrently.
>
> Regardings gradient clipping:
> It is standard to apply gradient clipping in finetuning and we also applied this technique in our experiments. We have updated the paper (page 3, experimental setup). to clarify.
>
> Finally, on MNLI, we do see faster model convergence at the beginning with bias-correction. The final performance is roughly the same though since bias-correction affects mostly the early updates (please see general reply for the numbers).

---

### Author Response · Authors · 2020-11-17
**Bias correction on a large dataset (MNLI)**

All reviewers suggest experimenting on MNLI, as an instance of a large dataset, to confirm if Adam with bias correction gets similar results, as we hypothesize in our work. Here are the numbers we get with three random runs:

| Setup                                    | Dev Accuracy (%) | Test Accuracy (%) |
|-------------------------------------|----------------| ------------------|
| No Bias Correction             | 86.0 +/- 0.3  | 87.0 +/- 0.4 |
| Bias Correction                   | 85.9 +/- 0.3  | 86.9 +/- 0.3 |

Without bias correction, the average accuracy is only 0.1% better. This gap is smaller than the standard deviation. This further confirms the hypothesis suggested by Figure 1 in the paper: there is no significant difference on large datasets. We have updated this information into the Appendix of the paper. Most important, there’s no drop in performance for larger datasets, and the same algorithms can be used for small and large datasets.

---

### Decision · Program_Chairs · 2021-01-07
**Final Decision**

**Decision:**

Accept (Poster)

**Comment:**

This paper addresses some of the well-documented instabilities that can arise from fine-tuning BERT on a dataset with few samples. Through a thorough investigation, they highlight various bizarre behaviors that have a negative impact on stability: First, that BERT inexplicably uses an unusual variant of Adam that, in fact, harms behavior; and second, that people tend to undertrain BERT on some downstream tasks. Separately, they find that reinitializing some of the final layers in BERT can be helpful. Since fine-tuning BERT has become such a common way to attack NLP problems, these practical recommendations will be quite welcome to the community. These findings address issues raised by recent work, so the paper is timely and relevant. The paper has through empirical analysis and is clear to read. There is a concurrent ICLR submission with similar findings, and this paper stands on its own. Reviewers all agreed that this paper should be published.